# Subnanometric Ru clusters with upshifted D band center improve performance for alkaline hydrogen evolution reaction

Qi Hu[1], Keru Gao[1], Xiaodeng Wang[2], Hongju Zheng[1], Jianyong Cao[1], Lingren Mi[1], Qihua Huo[1], Hengpan Yang[1], Jianhong Liu[1] & Chuanxin He [1]✉

Subnanometric metal clusters usually have unique electronic structures and may display electrocatalytic performance distinctive from single atoms (SAs) and larger nanoparticles (NPs). However, the electrocatalytic performance of clusters, especially the size-activity relationship at the sub-nanoscale, is largely unexplored. Here, we synthesize a series of Ru nanocrystals from single atoms, subnanometric clusters to larger nanoparticles, aiming at investigating the size-dependent activity of hydrogen evolution in alkaline media. It is found that the d band center of Ru downshifts in a nearly linear relationship with the increase of diameter, and the subnanometric Ru clusters with d band center closer to Femi level display a stronger water dissociation ability and thus superior hydrogen evolution activity than SAs and larger nanoparticles. Benefiting from the high metal utilization and strong water dissociation ability, the Ru clusters manifest an ultrahigh turnover frequency of $43.3\,s^{-1}$ at the overpotential of 100 mV, 36.1-fold larger than the commercial Pt/C.

[1] College of Chemistry and Environmental Engineering, Shenzhen University, Shenzhen, Guangdong 518060, PR China. [2] School of Electronic Information and Electrical Engineering, Chongqing University of Arts and Sciences, Chongqing 400030, PR China. ✉email: hecx@szu.edu.cn

Electrochemical water splitting is widely recognized as a promising approach for efficient, green, and sustainable hydrogen production[1–4]. In general, the water electrolysis can be operated in either acid or alkaline media. While the proton exchange membranes (PEMs)-enabled acid water electrolysis has demonstrated some advantages, its wide-spread application is still plagued by the high-cost of PEMs[5] and the sluggish kinetics of anodic oxygen evolution reaction (OER) in acid media[6]. The above issues can be alleviated, when the electrolysis is performed in alkaline media[7,8]. Nevertheless, this brings about a new disadvantage, that is the markedly obstructed kinetics of cathodic hydrogen evolution reaction (HER) in alkaline media[9,10]. Even on the benchmark Pt/C, the HER kinetics in alkaline is 2 orders magnitude inferior than that in acid[11]. Therefore, it is highly desirable to develop robust electrocatalysts for boosting the alkaline HER, achieving thus efficient alkaline water electrolysis.

Alkaline HER consists of three elementary steps[12]:

$$H_2O + e^- \rightarrow H_{ad} + OH^- \text{ (Volmer step).} \quad (1)$$

$$H_2O + H_{ad} + e^- \rightarrow H_2 + OH^- \text{ (Heyrovsky step).} \quad (2)$$

$$H_{ad} + H_{ad} \rightarrow H_2 \text{(Tafel step).} \quad (3)$$

The Volmer step, a unique step in alkaline HER that is absent in acid media, requires considerable energy to break the OH–H bond, resulting in the slower reaction rate of alkaline HER. To this end, enhancing the water dissociation ability has emerged as an efficient strategy for accelerating the alkaline HER activity[13,14]. Guided by this strategy, the alkaline HER activity of Pt has been improved by incorporating another component with strong water dissociation ability (i.e., metal oxides or hydroxides)[11,15,16]. Despite the enhanced activity, the mass usage of Pt in HER seems impractical due to the high cost and low reserves. Recently, Ru has attracted intensive interest as a cost-saving alternative to Pt due to the similar metal-hydrogen bond strength of both metals[17]. As such, a variety of highly active Ru-based electrocatalysts has been developed for HER, such as the carbon nanotubes supported ultrafine Ru nanoparticles (NPs) and Au–Ru nanowires[18,19]. To enhance the metal utilization efficiency and thus reduce the usage of Ru, Ru single atoms (SAs)-based electrocatalysts were also synthesized[20,21]. Whereas their high activity for acid HER, Ru SAs are not very active for alkaline HER, and the reported Ru SAs with good activity for alkaline HER usually also contain a large content of Ru NPs[5,22,23]. To explore fundamental reasons underlying this unusual phenomenon, Cho et al., performed systematic experimental and theoretical studies[24]. They found that Ru SAs have suitable binding energy towards H* intermediates, however, their weak ability of water dissociation greatly hinders the Volmer step and thus limits the overall reaction rate of alkaline HER. To this end, incorporating Ru NPs of good water dissociation ability nearby Ru SAs have proven to be a valid approach for significantly enhancing the alkaline HER activity of Ru SAs. This strategy seems facile yet challenging at the same time due to the significant difficulty in the synthesis of hybrids of NPs/SAs, especially the precise control over the distance between NPs and SAs. Moreover, introducing Ru NPs would reduce the metal utilization efficiency and thus lead to extra cost of using more Ru. Therefore, other strategies are urgently required to boost the water dissociation ability of Ru while ensuring high metal utilization.

Subnanometric metal clusters, with dimensions between metal SAs and NPs, have demonstrated unique electronic structures and thus extraordinary physical and chemical features distinctive from SAs and NPs[25–27]. So, one can expect that subnanometric clusters interact differently with reactants and intermediates, showing unusual selectivity and activity towards various catalytic reactions[28]. For example, Hutchings et al. found that subnanometric Au clusters (~0.5 nm) were highly active for CO oxidation, whereas the activity of Au SAs and large Au NPs (>5 nm) was very low[29]. Furthermore, subnanometric clusters with high surface-to-volume enable much higher metal utilization with respect to larger NPs[30]. Therefore, we conceived that the subnanometric Ru clusters with unique electronic structures may have a strong ability for water dissociation while allowing high metal utilization, thereby enabling low-cost and efficient HER in alkaline media. However, the HER activity of subnanometric metal clusters is largely unexplored and awaits further investigations.

Herein, we report the synthesis of subnanometric Ru clusters (i.e., 1 nm) and larger Ru nanoparticles (i.e., 2.3 and 3.1 nm) with the same mass loading on a carbon support, with the goal of studying the unique physicochemical features and electrocatalytic performance of Ru clusters distinctive from Ru nanoparticles. Based on both experimental and theoretical results, we discover that the diameter of Ru directly determines the energy level of the d band center, and Ru clusters with upshifted d center possess a stronger water dissociation ability than Ru SAs and larger Ru NPs for enabling much higher HER activity. Moreover, the Ru clusters with high surface-to-volume provide highly exposed active sites to significantly improve the metal utilization efficiency of Ru. Consequently, the Ru clusters display excellent performance for alkaline HER with a small overpotential of 13 mV at 10 mA cm$^{-2}$ and ultrahigh TOF of 43.3 s$^{-1}$ at an overpotential of 100 mV, surpassing other reported electrocatalysts. Impressively, when the mass loading increases to 1 mg cm$^{-2}$, the Ru cluster can also sustainably enable a large current density of 1000 mA cm$^{-2}$ at the overpotential of 196 mV. Our results thus unveil the exceptional electrocatalytic performance of subnanometric metal clusters and the fundamental origin underlying the performance.

## Results

In general, the size of metal nanoparticles greatly impacts their electronic structures (i.e., d band center), which may in turn alter the HER performance[31]. However, this still remains elusive due to the lack of systematic investigation. To this end, density functional theory (DFT) calculations were first performed to study the electronic structures alternation of Ru clusters induced by the different particle sizes. We selected odd-number of Ru$_n$ clusters ($n$ = 19, 55, and 79) to investigate the size effect of Ru on HER activity because they possess singly occupied highest occupied molecular orbitals (HOMO), rendering them more active than the even-numbered Ru clusters with a doubly occupied HOMO[32]. As displayed in Fig. 1a and Supplementary Fig. 1, with increasing the particle size of Ru clusters from Ru$_{19}$ to Ru$_{79}$ (19 and 79 represent the atomic number of Ru clusters), the d band center of Ru gradually downshifts in a nearly linear way. This can be attributed to the decreased proportion of low-coordinated Ru atoms with increasing the particle size of Ru clusters[33]. In other words, the smallest Ru$_{19}$ clusters with the largest proportion of low-coordinated Ru atoms display the d band center closest to the Femi level. Notably, the adsorption strength of Ru clusters towards both H$_2$O reactants and H* intermediates decreases also in a nearly linear way with the downshifts of the d band center (Fig. 1b, c). This can be explained by the electron-interaction model, in which bonding and antibonding states are generated between the adsorbate valence p-level and metal d-band[34,35]. The antibonding states will result in the Pauli repulsion and consequently reduce the bond strength. The downshift of the d band center will make more antibonding states below the Fermi level, leading to weakened bond strength. Thus, the linear upshift of the d band center with reducing the particle size of Ru clusters enables the enhanced adsorption strength of H$_2$O and *H. Accordingly, the smallest Ru$_{19}$ cluster has

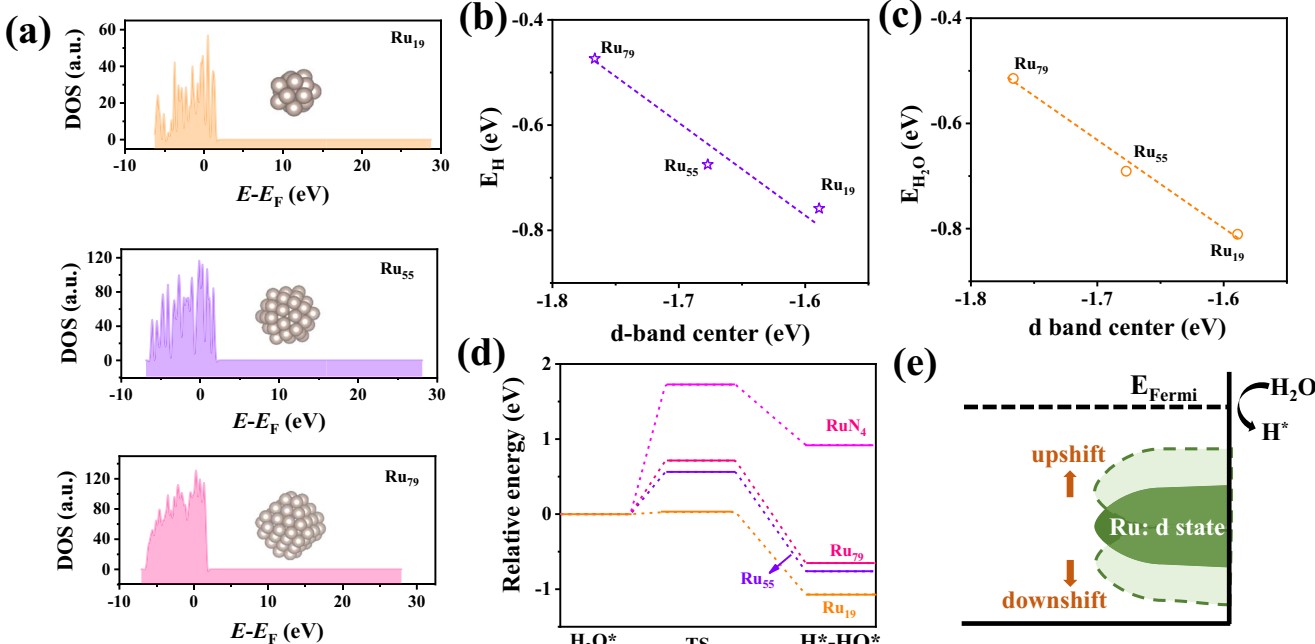

**Fig. 1 Theoretical calculations of d band center for different Ru clusters. a** Projected electronic densities of states (pDOS) for $Ru_{19}$, $Ru_{55}$, and $Ru_{79}$ cluster models. Correlation between (**b**) $E_{H2O}$, **c** $E_H$ and d band center on different Ru cluster models. **d** Kinetic barriers of water dissociation for different Ru cluster models. **e** Schematic illustrating that the d band center shift of clusters induced by the particle size tunes the water dissociation ability.

the strongest adsorption strength towards both $H_2O$ and *H. The strong adsorption strength of $H_2O$ on $Ru_{19}$ clusters may make the H–O bond of adsorbed $H_2O$ to be elongated for facilitating the dissociation of $H_2O$ and markedly accelerating the sluggish Volmer step of alkaline HER[36]. As expected, the $Ru_{19}$ cluster is highly active for the water dissociation with the energy barrier of merely 0.03 eV. (Fig. 1d) In sharp contrast, the energy barrier of $Ru_{55}$ and $Ru_{79}$ are as large as 0.56 and 0.71 eV for the water dissociation, respectively, signifying that both the two clusters suffer from a poor ability for breaking the H–OH bond, which impedes the generation of H* and results in sluggish reaction kinetics of alkaline HER. Furthermore, we also investigated the water dissociation ability of Ru SAs (i.e., $RuN_4$). In consistent with the previous literature[24], the $RuN_4$ is inert for the water dissociation with a significant energy barrier of 1.71 eV (Fig. 1d), and this is the reason why the Ru SAs are not very active for the alkaline HER. Optimized surface configurations for adsorption of H* and water dissociation on different structural models are shown in Supplementary Figs. 2–6. Overall, the $Ru_{19}$ cluster with upshifted d band center has a much stronger water dissociation ability than Ru SAs and larger clusters (i.e., $Ru_{55}$ and $Ru_{79}$) (Fig. 1e), rendering the $Ru_{19}$ cluster be highly active for the alkaline HER. However, to the best of our knowledge, the HER activity of subnanometeric Ru clusters (i.e., $Ru_{19}$) has yet been reported and awaits further investigations.

As a proof-of-concept study, we synthesized subnanometric Ru clusters and larger Ru NPs loaded on a porous carbon substrate through the calcination in $H_2$/Ar flow at different temperatures (details see Experimental section). The resulting samples were denoted as Ru-x, where x stands for the average diameter of Ru species. In order to eliminate the influence of Ru content on the electrocatalytic performance, all samples were prepared using the same amount of Ru source, and the mass loading of Ru was determined by using inductively coupled plasma-atomic emission spectrometry (ICP-AES). The results indicate that the Ru content in all three samples is quite similar (~2.0 wt%), and only a slight decrease in Ru content can be observed with the increase of calcination temperature. (Supplementary Fig. 7) As displayed in

Fig. 2a–c and Supplementary Figs. 8–10, transmission electron microscopy (TEM) images manifest uniformly dispersed Ru nanocrystals in all three samples without obvious aggregation. Corresponding size-distribution histograms based on 150 nanocrystals provide average size of 1.0, 2.3, and 3.1 nm, respectively (Fig. 2d–f). Intriguingly, with the increase of calcination temperature from 300, 400 to 500 °C, the diameter of Ru crystals increases accordingly from 1, 2.3, to 3.1 nm. This is because the higher calcination temperature accelerates the movement of Ru atoms during the calcination and thus leads to the growth of Ru crystals. Atomic-scale structure of Ru nanocrystals in Ru-1.0 was further scrutinized by using the high-angle annular dark-field scanning transmission electron microscopy (HAADF-STEM) (Fig. 2g). One can see several bright speckles with a diameter of ~1.0 nm corresponding to Ru clusters. Moreover, the Ru clusters are amorphous and composed of ~20 atoms. Elemental mappings based on the HADDF-STEM images further confirm the successful synthesis of Ru clusters supported on the carbon substrate. (Supplementary Fig. 11) X-ray diffraction (XRD) patterns of all three samples only display peaks for graphitized carbon without those for metallic Ru (Supplementary Fig. 12), indicating that the Ru nanocrystals are amorphous in all the three samples. In addition, after loading Ru clusters, the surface area of carbon substrate decreases from 1890 to 1630 $m^2 g^{-1}$, and the intensity of micropore distribution is reduced (Supplementary Fig. 13), implying that the spatial confinement of micropores may be essential for the formation of subnanometric clusters in the Ru-1.0. If the carbon substrate is not porous, larger Ru nanoparticles are generated (Supplementary Fig. 14), further confirming the important role of micropores in generating Ru clusters. For comparison, we also synthesized Ru SAs (i.e., $RuN_4$) on the same porous carbon substrate by adding melamine and decreasing the introduced amount of $Ru^{3+}$ (The detailed synthesis process, please see "Experimental Section") (Supplementary Fig. 15).

Electronic structures and work function values (Φ) of Ru in the above three samples were investigated through ultraviolet photoelectron spectroscopy (UPS). The calculated Φ values are 3.32,

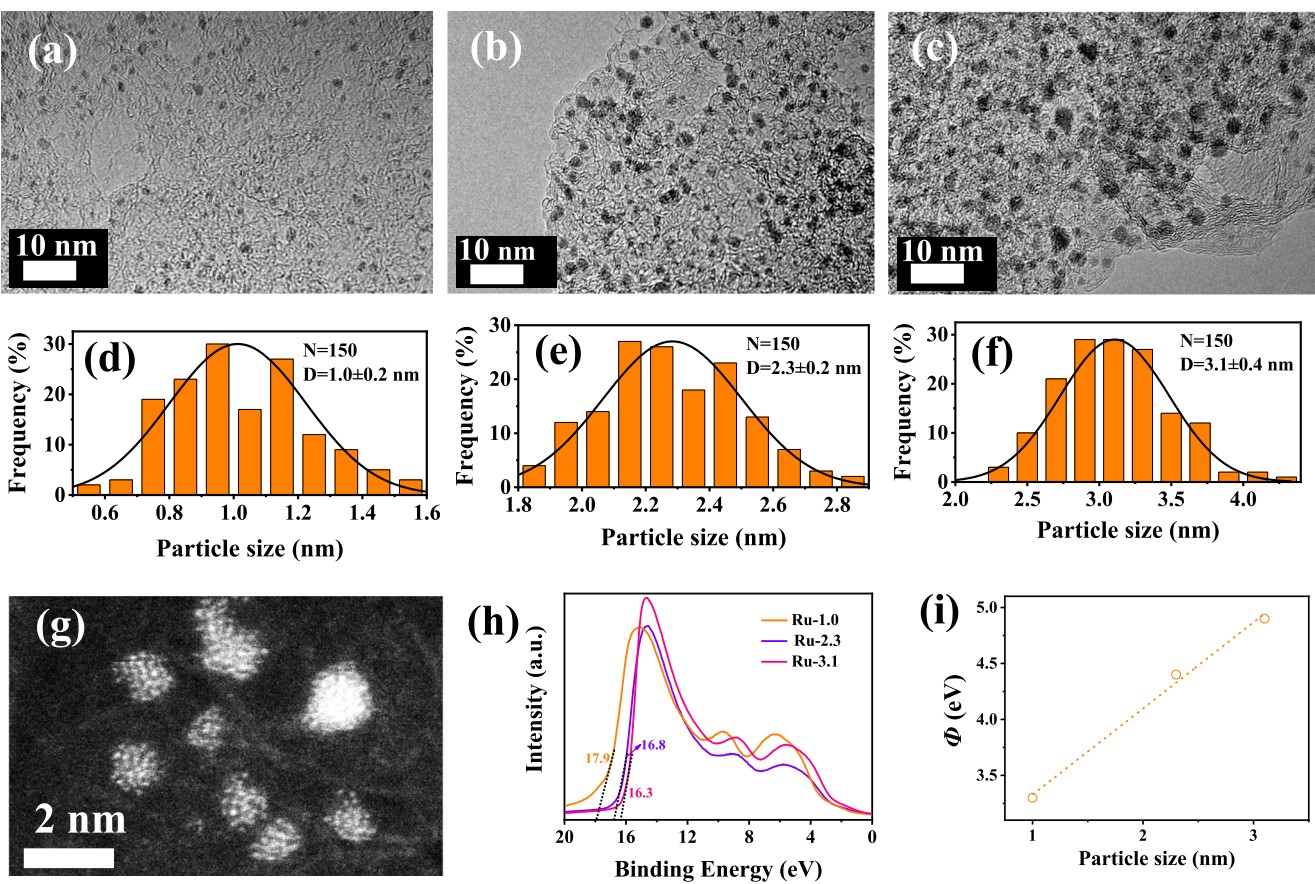

**Fig. 2 Particle size and work function values of various Ru-based samples.** TEM images of Ru crystals in different size: **a** Ru-1.0, **b** Ru-2.3, **c** Ru-3.1. **d–f** Corresponding size distribution of various Ru samples. **g** HADDF-STEM images of Ru-1.0. **h** UPS spectra of Ru clusters. **i** Correlation between particle size and Φ values.

4.42, and 4.92 eV for Ru-1.0, Ru-2.3, and Ru-3.1, respectively. (Fig. 2h) Notably, with the increase in diameter of Ru nanocrystals, the Φ value also increases in a nearly linear way (Fig. 2i), indicating that Φ values of Ru can be efficiently tuned by the change of particle diameter. As shown in Supplementary Fig. 16, the smaller Φ value suggests a higher Fermi energy level ($E_F$)[37], thus illustrating that the d band center ($E_d$) energy level follows the order that Ru-1.0 > Ru-2.3 > Ru-3.1. This is in good consistence with the above DFT calculations that the d band center of Ru gradually downshifts with the increase of particle size. The smaller Φ value of Ru-1.0 also implies the smaller energy barrier of electron transfer from the surface of electrocatalysts to reactants and intermediates[38,39], and this also favors the HER. The electronic structures of Ru were further studied by the X-ray photoelectron spectroscopy (XPS) spectra. With the particle size increasing from 1.0 to 3.1 nm, the binding energy of Ru $3p_{3/2}$ shifts from 462.8 to 463.3 eV (Supplementary Fig. 17), indicating that the Ru-1.0 has a higher electron density for upshifting the d band center. Moreover, we also determined the Φ value of Ru SAs through UPS spectra. (Supplementary Fig. 18) It can be found that the Φ value of Ru SAs (3.12 eV) is smaller than the Ru-1.0 (3.32 eV), implying that the higher Fermi energy level of Ru SAs than the Ru-1.0. This further confirms the fact that the d band center of Ru upshifts with the decrease of Ru particle size. With upshifted d band center, the Ru-0.1 is expected to display excellent performance for the HER in alkaline media.

The HER performance of Ru-based electrocatalysts (i.e., Ru-1.0, Ru-2.3, and Ru-3.1) and commercial Pt/C was evaluated in 1 M KOH. Linear sweep voltammetric (LSV) curves of the above four electrocatalysts are shown in Fig. 3a, indicating that the

Ru-1.0 possesses the best HER performance with the highest current density at each tested overpotential. As displayed in Fig. 3b, the Ru-1.0 requires an overpotential of merely 13 mV to yield the current density of 10 mA cm$^{-2}$, and such overpotential is 9 and 11 mV-negative than that on the Ru-2.3 and Ru-3.1. Moreover, the value of 13 mV is also much lower than that of the Pt/C (38 mV), further confirming the good HER performance of Ru-1.0. The HER activity of different electrocatalysts is further compared by the overpotential at a large current density of 100 mA cm$^{-2}$, which is required by the practical water electrolysis. At 100 mA cm$^{-2}$, the Ru-1.0 requires an overpotential of 60 mV, 31- and 38 mV smaller than that of Ru-2.3 and Ru-3.1 (Fig. 3b). Moreover, the overpotential gap between Ru-1.0 and the other two Ru-based electrocatalysts increases with the increase of current density. (Supplementary Fig. 19) This can be attributed to the much stronger water dissociation ability of Ru-1.0 than the other two electrocatalysts, as demonstrated in CO stripping experiments, discussed later. At a large current density (i.e., >100 mA cm$^{-2}$), H* intermediates require to be rapidly generated via the water dissociation for increasing the coverage of H* and thus enabling the dimerization of H* at a high rate. Therefore, the Ru-1.0 with strong water dissociation ability can increase the H* coverage at the surface of electrocatalysts to allow rapid hydrogen evolution for achieving large current densities at small overpotentials.

As we previously reported[40], the porous carbon derived from the litchi shell contains a small amount of nitrogen (N) (1.1 wt%). The N elements in carbon may electronically interact with Ru clusters to promote their catalytic performance[41]. To this end, we synthesized porous carbon nanomaterials without nitrogen by

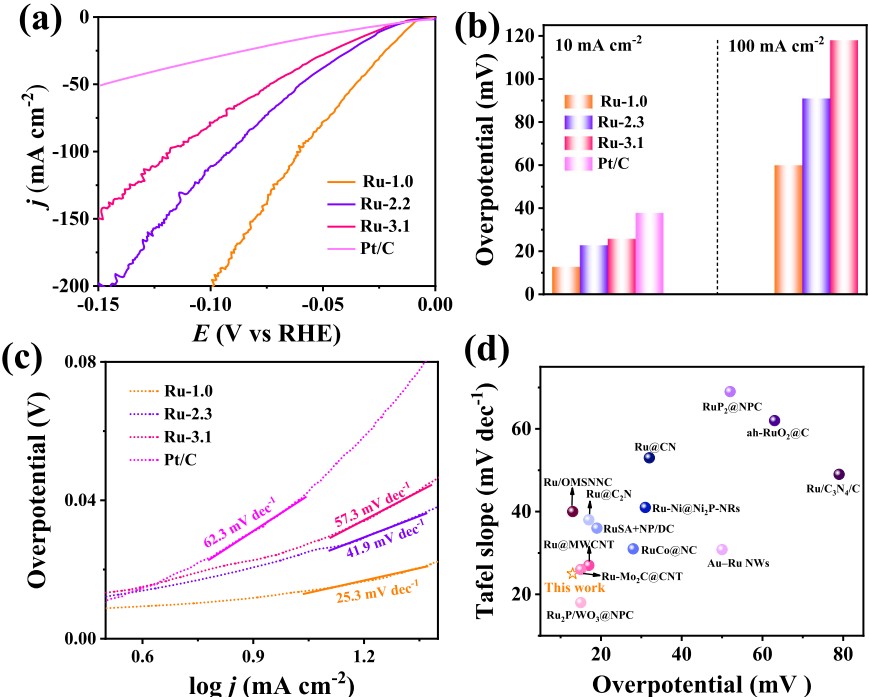

**Fig. 3 HER performance of different Ru-based electrocatalysts. a** LSV curves of Ru-1.0, Ru-2.3, Ru-3.1, and commercial Pt/C for HER in 1 M KOH. **b** Comparing the overpotentials of above noted samples at the current density of 10 and 100 mA cm$^{-2}$. **c** Tafel slopes of above samples. **d** Comparing HER performance of our Ru-1.0 with other reported electrocatalysts on the basis of overpotential at 10 mA cm$^{-2}$ and Tafel slopes.

using potassium citrate as precursors (detailed synthetic process, please see Experimental section), and then we employed this porous carbon as substrates to obtain subnanometric Ru clusters in diameter of ~1 nm. The resulting sample was denoted Ru-1.0/N-free C, and corresponding HRTEM images of Ru-1.0/N-free C further confirmed that the Ru clusters were in diameter of ~1 nm. (Supplementary Fig. 20) As displayed in Supplementary Fig. 21, the HER activity of Ru-1.0/N-free C is inferior to the Ru-1.0 (supported on the porous N-doped carbon), suggesting that the nitrogen on the carbon may induce the electronic interaction between Ru clusters and carbon support for promoting the HER. The N-induced metal/support interactions were further investigated by performing DFT calculations using the structural models of Ru$_{19}$ clusters supported on the N-doped carbon (denoted Ru$_{19}$/NC) (Supplementary Fig. 22). The electronic density difference diagram indicates the presence of electron transfer from Ru$_{19}$ clusters (yellow region stands for the electron loss) to N-doped carbon (cyan region stands for the electron accumulation) at interfaces, thereby tuning the electronic structure of Ru (Supplementary Fig. 23). Bader charge analysis further suggests that the number of transferred electrons from Ru clusters to the nitrogen-doped carbon is 1.48|e|. Notably, compared with pure Ru$_{19}$ (i.e., without the N-doped carbon support), the binding energy of the H* intermediate is decreased on the Ru$_{19}$/NC (Supplementary Fig. 24), which in turn facilitates the desorption of H* for boosting the HER. The optimized structural model of H* adsorbed on the Ru$_{19}$/NC is shown in Supplementary Fig. 25. Moreover, the adsorption of Ru$_{19}$ clusters on N-doped carbon is a highly exothermic process with an adsorption energy of −8.15 eV, suggesting that the N-induced metal/support interaction can enhance the stability of Ru clusters during HER. Therefore, the N-induced metal/support interaction also contributes to the excellent HER activity of Ru-1.0 by optimizing the electronic structure of Ru and enhancing the stability of Ru.

Tafel slope is a good indicator of reaction kinetics and the rate-determining step (RDS) of electrocatalytic reactions. In general,

different values of the Tafel slope indicate different reaction processes and RDS. As displayed in Fig. 3c, the Pt/C enables a Tafel slope value of 62.3 mV dec$^{-1}$, implying that the HER undergoes the Volmer–Heyrovsky route with the RDS of the Heyrovsky step (Supplementary Fig. 26)[42,43]. It should be noted that the Heyrovsky step also involves the water dissociation, and thus the reaction kinetics of HER is still limited by the sluggish water dissociation. With Tafel slope values greater than the 30 mV dec$^{-1}$, the Ru-2.3 and Ru-3.1 also follow the route of Volmer–Heyrovsky. In striking contrast, the Tafel slope value of Ru-1.0 is only 25.3 mV dec$^{-1}$, smaller than the value of 30 mV dec$^{-1}$, suggesting that the HER undergoes the route of Volmer-Tafel with the RDS of Tafel step. As the Tafel step only involves the dimerization of H* without the water dissociation, the RDS of HER on the Ru-1.0 is not associated with the water dissociation. In other words, the water dissociation is significantly promoted on the Ru-1.0. Furthermore, the smallest Tafel slope value of Ru-1.0 indicates the most rapid reaction kinetics of Ru-1.0 for HER, in consistence with the results demonstrated on LSV curves. The electrochemical impedance spectroscopy (EIS) manifests the smallest charge transfer resistance (R$_{ct}$) of Ru-1.0 (Supplementary Fig. 27), providing further evidence for the most rapid HER kinetics of Ru-1.0. Thus, the high activity and reaction kinetics of HER on the Ru-1.0 should have a great relationship with the enhanced ability of water dissociation.

Long-time durability is very important for the practical application of an HER electrocatalyst. After 1000 cycles of CV, no obvious change is observed on the LSV curves of Ru-1.0 (Supplementary Fig. 28), indicating the high stability of the HER in alkaline media. To further demonstrate the high activity of Ru-1.0, we compare the HER activity of Ru-1.0 with other reported Ru-based electrocatalysts (Fig. 3d and Supplementary Table 1), indicating that the Ru-1.0 is superior to most reported electrocatalysts.

Systematic investigations combing both DFT calculations and various electrochemical tests have demonstrated that the water

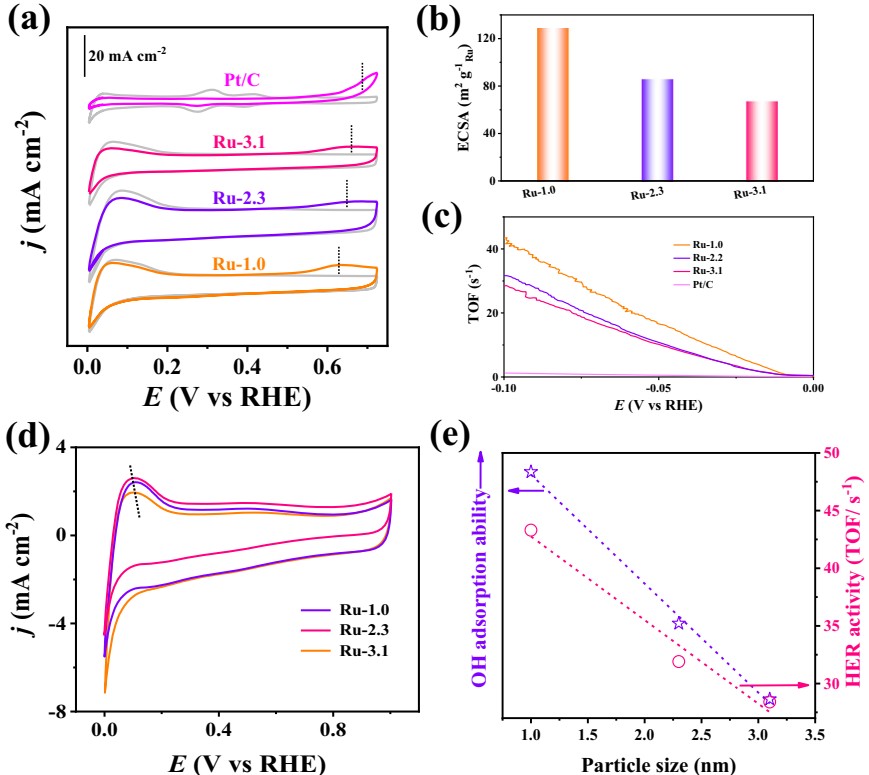

**Fig. 4 Relationship between HER performance and water dissociation ability. a** CO stripping voltammetry of different Ru samples and commercial Pt/C. The dashed lines stand for the CO oxidation potentials. **b** Comparing ECSA values of Ru samples. **c** Potential-dependent TOF curves of above noted samples. **d** CV curves of different Ru samples at the potential range of 0-1.0 V vs. RHE. **e** Relationship of OH adsorption ability, HER activity, and particle size of different Ru samples.

dissociation ability is a key descriptor for different HER activity on our Ru-based electrocatalysts with different particle diameters. As demonstrated by Qiao's group, the $OH^-$ concentration greatly impacted the HER activity of Pt/C electrocatalyst, and a high $OH^-$ concentration could significantly boost the water dissociation to provide a unique acid-like local reaction environment for enhancing HER activity. Thus, the adsorption ability of *OH is a key parameter to evaluate the water dissociation ability of an electrocatalyst, and the water dissociation ability of different electrocatalysts can be determined by the *OH bind energy (OHBE)[44,45] Since *OH can promote the removal of surface adsorbed *CO intermediates, we performed CO stripping experiments to evaluate the OHBE of different electrocatalysts[46]. The result of Pt/C is shown in Fig. 4a, revealing a CO-stripping peak situated at 0.69 V, in accordance with previous literature[46]. CO-stripping peaks at lower potentials are observed on the Ru-2.3 (0.65 V) and Ru-3.1 (0.66 V), indicating their higher OHBE than the Pt/C. Notably, the Ru-1.0 displays the lowest potential of 0.63 V for the CO-tripping peak, suggesting the highest OHBE and thus the best water dissociation ability. This is in line with the above DFT calculations and UPS results that the upshifted d band center of Ru in the Ru-1.0 enhances the adsorption of OH* for promoting the dissociation of water.

The results of CO-stripping were further employed to determine the electrochemically active surface area (ECSA) values. As expected, the Ru-1.0 has a larger ECSA value than the Ru-2.3 and Ru-3.1 (Fig. 4b), indicating the higher metal utilization efficiency of subnanometric metal clusters than the larger NPs. To eliminate the contributions of ECSA to the tested HER activity, we calculated the turnover frequency (TOF) values of different electrocatalysts on the basis of ECSA values (Fig. 4c). At the overpotential of 10 and 100 mV, the Ru-1.0 delivers larger TOF values of 1.2 and 43.3 $s^{-1}$, respectively, much larger than those of Ru-2.3 (0.7 and 31.9 $s^{-1}$)

and Ru-3.1 (0.8 and 28.4 $s^{-1}$), suggesting the higher intrinsic HER activity of Ru-1.0. Impressively, at the overpotential of 100 mV, the TOF value of Ru-1.0 is 36.1-fold larger than the commercial Pt/C, and also superior to the other reported electrocatalysts (Supplementary Table 2). Besides, we also compared the HER activity of Ru-1.0 and SAs Ru on the basis of TOF values. (Supplementary Fig. 29) In consistence with the above DFT calculations, the SAs Ru with poor water dissociation ability demonstrates much lower HER activity than the Ru-1.0. Intriguingly, the gap of TOF values between Ru clusters and SAs Ru increase with the increase of overpotentials. This can be due to the stronger dissociation ability of Ru clusters for increasing the H* coverage at the electrocatalyst surface, thereby achieving high TOF values at small overpotentials. To further confirm that the subnanometric cluster can enable high metal utilization efficiency, we also evaluated the ECSA of different electrocatalysts through the double-layer capacitance ($C_{dl}$) values obtained from cyclic voltammetry (CV) curves over a Faradic current density-free range. (Supplementary Fig. 30) The $C_{dl}$ value Ru-1.0 is 117.5 mF $cm^{-2}$, larger than the Ru-2.3 (96.9 mF $cm^{-2}$) and Ru-3.1 (82.8 mF $cm^{-2}$), further confirming the subnanometric clusters indeed promote the exposure of active sites.

As another activity descriptor for HER electrocatalysis, hydrogen binding energy (HBE) also greatly impacts the HER activity in alkaline media. To this end, we experimentally measured HBE values of different electrocatalysts using a method of CV at a potential range of 0–1 V vs. RHE in Ar-saturated 1 M KOH. Corresponding CV curves manifest typical patterns containing two regions: adsorption/desorption peaks of underpotentially deposited hydrogen ($H_{upd}$) at the potential range of 0–0.2 V (vs. RHE) and a double-layer potential range from 0.2–1.0 V (vs. RHE). As reported, the potential of the $H_{upd}$ desorption peak ($E_{peak}$) has a direct relationship with the HBE of active sites by the equation:

$\Delta H = -FE_{peak}$[47,48]. According to this equation, we calculated the HBE values of different electrocatalysts, indicating that the Ru-1.0 has a higher HBE value than the Ru-2.3 and Ru-1.0. (Fig. 4d) This is consistent with the results of UPS and DFT calculations that the upshifted d center of Ru in the Ru-1.0 decreases the filling of metal-H antibonding states above the Fermi lever, leading to a relatively stronger H adsorption strength of Ru-1.0 with respect to the Ru-2.3 and Ru-3.1. Indeed, the too strong adsorption strength of H on the Ru-1.0 should be adverse to HER for preventing the desorption of H, and it seems contradictory with the fact that the Ru-1.0 has the best HER activity. To address this contradiction, we correlated the OHBE and HBE values with the alkaline HER activity of three electrocatalysts. As displayed in Fig. 4e, the HER activity increases monotonically with the increase of the HOBE value. In other words, the HER activity increases monotonically with the improvement of water dissociation ability. However, as for the HBE, the Ru-3.1 with a more suitable HBE value (closer to the optimum value of 0) displays the inferior HER activity than the other two electrocatalysts. (Supplementary Fig. 31) The above results reflect that the water dissociation ability is the governing factor to determine the HER activity of our Ru-based electrocatalysts, and the subnanometric Ru clusters with upshifted d band center have stronger water dissociation ability for enabling superior HER activity than larger Ru NPs.

HER activity of different electrocatalysts was also evaluated in the acid electrolyte (i.e., 0.05 M $H_2SO_4$ solution). (Supplementary Fig. 32) Surprisingly, the HER activity trend in the acid electrolyte is totally opposite to that in alkaline, and the Ru-1.0 displayed the inferior HER activity than that of Ru-2.3 and Ru-3.1 in 0.05 M $H_2SO_4$. It is widely accepted that the HER activity in the acid electrolyte is only determined by hydrogen binding energy (HBE) of the electrocatalyst surface, while the alkaline HER activity is jointly determined by both water dissociation ability and (HBE)[11,49,50]. In this work, DFT calculations and experiment results have demonstrated that the Ru-1.0 with upshifted d band center has too strong H adsorption strength and thus suppresses the desorption of H, compared with the other two electrocatalysts. In this regard, the inferior HER activity in acid electrolytes than the other two electrocatalysts can be attributed to the too strong H adsorption strength. As for the alkaline HER, the water dissociation ability is a governing factor to control the HER activity, and the Ru-1.0 with stronger water dissociation ability displays superior HER activity than the other two electrocatalysts. The above findings further confirm that the superior HER activity of Ru-1.0 in alkaline is owing to the strong water dissociation ability.

To optimize the HBE value and further enhance the HER activity of Ru-1.0, we attempted to tune the electronic structure of Ru by doping another element of Co. LSV curves of Ru-1.0 with different content of Co are shown in Supplementary Fig. 33, indicating that doping a small amount of Co (i.e., molar ration of Ru/Co = 4) can increase the HER activity. XPS spectra show that the Co dopants modify the electronic structure of Ru (Supplementary Fig. 34). Furthermore, CV curves suggest that the adsorption strength of H is weakened after doping Co (Supplementary Fig. 35). Thus, Co dopants optimize the electronic structures of Ru and thus reduce the adsorption strength of H for favoring HER. This also implies that the adsorption strength of H is also important for efficient HER in alkaline media.

In order to demonstrate that the Ru-1.0 can be synthesized on a large scale, the added amount of raw materials was increased by 37 times, and the obtained product could be up to 2.0 g in one batch, with a yield of 90.5%. (Supplementary Fig. 36) The carbon support is low-cost and derived from biomass wastes (i.e., litchi pericarp). Moreover, the synthetic process is facile and only involves two steps of impregnation and subsequent calcination, providing great opportunities for the high throughput synthesis.

Besides, the Ru-1.0 was also loaded on the carbon paper with a mass loading of 1 mg cm$^{-2}$ to investigate the possibility of Ru-1.0 as electrocatalysts for high-output industrial $H_2$ production. In the electrolyte of 1 M KOH, the Ru-1.0 yields the large current densities of 500 and 1000 mA cm$^{-2}$ at small overpotentials of 119 and 196 mV, respectively (Fig. 5a). The overpotential of 119 mV at 500 mA cm$^{-2}$ is much smaller than that of commercial Pt/C (298 mV). Notably, the Ru-1.0 also compares favorably with other reported electrocatalysts at the current density of 1000 mA cm$^{-2}$. (Supplementary Table 3) The excellent performance of Ru-1.0 at large current densities may be attributed to the following two factors: 1) the strong water dissociation ability of Ru-1.0 enables rapid water dissociation for increasing the coverage of H* and thus promoting the dimerization of H* at a high rate; 2) the porous carbon support with good hydrophilicity (Supplementary Fig. 37) and high surface area (i.e., 1890 m$^2$ g$^{-1}$) significantly accelerates the mass transport during HER.

Furthermore, the Faradic efficiency (FE) of Ru-1.0 at 1000 mA cm$^{-2}$ was evaluated, and the obtained FE value of 98% provided further evidence that the Ru-1.0 was highly active for the HER (Fig. 5b). Long-time durability at large current densities (i.e., 1000 mA cm$^{-2}$) is vital for the practical application of an HER electrocatalyst. Accordingly, we tested the stability of Ru-1.0 at 1000 mA cm$^{-2}$ through a method of chronoamperometry (Fig. 5c). The fact that no current density decay is observed during the 100-h continuous HER test suggests that the Ru-1.0 has good stability for the large current density of HER. The good stability of Ru-1.0 can be further verified by the nearly unchanged particle size of Ru clusters and ECSA values (Supplementary Figs. 38 and 39). With small overpotentials and good stability at large current densities (i.e., 1000 mA cm$^{-2}$), the Ru-1.0 is a promising electrocatalyst for practical water splitting.

## Disscussion

In summary, we have demonstrated that subnanometric Ru clusters have electronic structures and alkaline HER performance distinctive from single Ru atoms and larger Ru nanoparticles. Specific, the Ru clusters manifest an ultrahigh TOF value of 43.3 s$^{-1}$ at the overpotential of 100 mV that markedly outperforms commercial Pt/C. To investigate the fundamental origin underlying the exceptional performance, we establish the connection between the diameter of Ru and the energy level of the d band center, suggesting that the d band center of Ru upshifts with the increase of diameter in a nearly linear way. Consequently, Ru clusters with the upshifted d band center have stronger water dissociation ability than SAs Ru and larger Ru nanoparticles for enabling higher HER activity. Impressively, the Ru clusters can be synthesized on a large scale and achieve a large current density of 1000 mA cm$^{-2}$ at a small overpotential of 196 mV with long-time stability, signifying that Ru clusters are promising for practical HER. The excellent electrocatalytic performance and unique physicochemical features of Ru clusters here open new doors to developing robust electrocatalysts by tuning the size of metal at the sub-nanoscale.

## Methods

**Electrocatalysts synthesis.** The porous carbon substrate with abundant micropores was prepared through the calcination of biomass-derived litchi shell in Ar with the use of $K_2CO_3$ as an etchant[40]. The Ru-x samples were then synthesized by adsorbing Ru$^{3+}$ cations on the porous carbon substrate, followed by the calcination in 10% $H_2$/Ar mixed gas. In a typic process, porous carbon (60 mg) and RuCl$_3$·$H_2$O (20 mg) were well dispersed in 5 ml deionized water under sonicated bath for 30 min to obtain a homogenous black suspension. And then, the suspension was centrifuged to obtain precipitate, which was washed with deionized water for three times. After the precipitate was dried at 60 °C for 24 h, the resulting powder was calcinated at different temperatures (i.e., 300, 400, and 500 °C) for 2 h in 10% $H_2$/Ar mixed gas. The obtained products were denoted Ru-1.0, Ru-2.3, and Ru-3.1, respectively, according to their particle diameter.

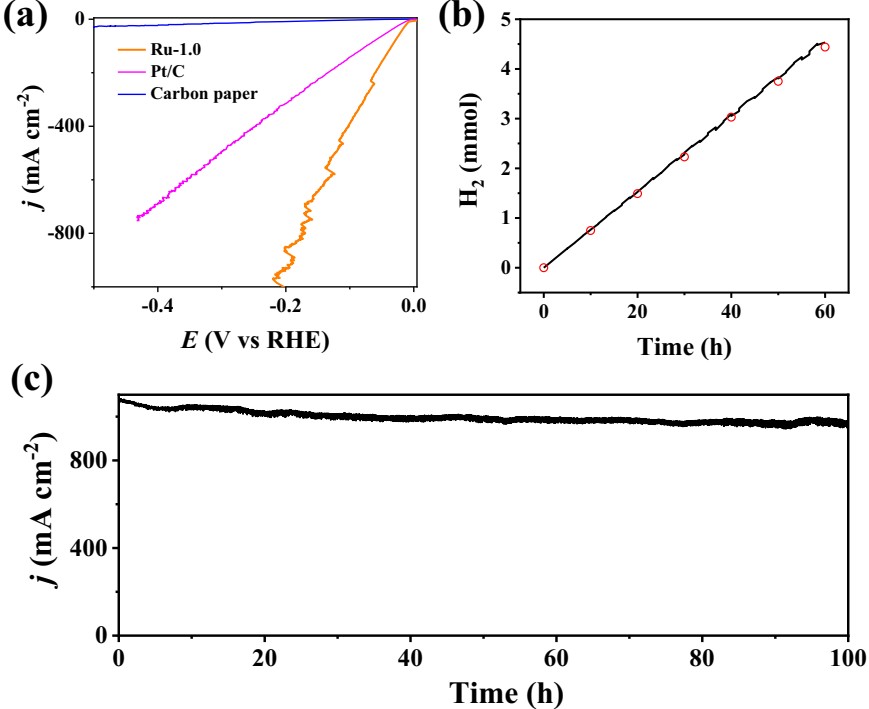

**Fig. 5 HER performance and stability of subnanometric Ru clusters at large current densities. a** LSV curves of Ru-1.0, commercial Pt/C and carbon paper for HER in 1 M KOH. The Ru-1.0 and commercial Pt/C are supported on the carbon paper with a high mass loading of 1 mg cm$^{-2}$. **b** Theoretical and experimental results of H$_2$ generation on the Ru-1.0. **c** Stability test of Ru-1.0 at the current density of 1000 mA cm$^{-2}$ for 100 h.

The Ru single atoms (SAs) supported on the porous carbon substrate were prepared according to the previous literature[51]. At first, the Ru$^{3+}$ cations were adsorbed on the porous carbon substrate through the same synthetic process of Ru-1.0 except that the introduced amount of RuCl$_3 \cdot$H$_2$O was halved. And then, the obtained powder was homogeneously mixed with melamine at a mass ratio of 1:5. Finally, the mixed powder was calcinated at 700 °C for 2 h in Ar gas, and the resulting sample was denoted Ru SAs.

In order to investigate the influence of nitrogen species on the HER activity, we also synthesized the porous carbon substrate without nitrogen by using potassium citrate as precursors, according to the previous literature[52]. Afterwards, subnanometric Ru clusters were loaded on the porous carbon substrate without nitrogen through the similar synthetic procedure as that of Ru-1.0.

**Characterizations**. XRD patterns of Ru-x samples were measured on the D8ADVANCE diffractometer equipped with a Kα radiation. The morphologies and particle size of different Ru crystals were investigated by high-angle annular dark-field scanning transmission electron microscopy (HADDF-STEM) images on a JEOL-2100F FETEM. XPS spectra were conducted over a Thermo VG ESCA-LAB250 X-ray photoelectron spectrometer. UPS measurements were performed on a UV/VIS spectrometer Lambada 25. The mass loadings of Ru on the porous carbon substrate were measured by ICP-AES (Shimadzu ICPS-7500 spectrometer). BET surface area was determined through low-temperature N$_2$ adsorption-desorption experiments on a Micromeritics ASAP 2020 sorptometer.

**Electrochemical measurements**. The HER performance of Ru-x and commercial Pt/C was evaluated at room temperature by using a typic three-electrode system connected to an electrochemical workstation (Autolab M204) in the electrolyte of Ar-saturated 1 M KOH. In the three-electrode system, a glassy carbon electrode (0.07 cm$^2$) with electrocatalysts (mass loading: 0.4 mg cm$^{-2}$) served as the working electrode, and Hg/HgO (1 M KOH) and graphite electrode were employed as the reference and counter electrode, respectively. Electrocatalysts inks were prepared by dispersing a certain amount of electrocatalysts in a mixed solution of deionized water (150 μL), ethanol (49 μL), and Nafion (9 μL) under a sonication bath. All the LSV curves for HER were recorded at a scan rate of 5 mV s$^{-1}$ after being calibrated 80% IR-compensations. Before recording LSV, five cycles of CV were performed at the potential range of $-0.4$–0 V vs RHE to activate electrocatalysts. EIS was conducted at an overpotential of 20 mV in a frequency range from 0.1 to 10$^5$ Hz. As for the CO stripping experiments, ultrapure CO (99.99%) gas was first adsorbed on the surface of electrocatalysts at a potential of 0.1 V vs RHE in the electrolyte of CO-saturated 1.0 M KOH for 10 min. Subsequently, ultrapure Ar (99.99%) was purged for another 10 min to remove the excess CO from the electrolyte. Afterwards, CV curves for CO stripping were recorded at a scan rate of 20 mV s$^{-1}$. According to the results of CO stripping,

ECSA values of Ru-x were determined by assuming that the electrooxidation of one CO* monolayer required a charge density of 420 μC cm$_{Ru}^{-2}$ [48].

**DFT calculations**. All density functional theory (DFT) calculations were conducted via the Vienna ab initio simulation package (VASP)[53]. The generalized gradient approximation with the Perdew, Burke, and Ernzerh of the exchange-correlation functional was used for the electron exchange and correlation energy[54]. For Kohn–Sham wave functions, the cutoff energy of the corresponding plane-wave basis set was set to 400 eV. The convergence criterion for the electronic self-consistent iteration (10$^{-4}$ eV) and the force for atomic relaxation (0.02 eV Å$^{-1}$) were employed, respectively. The adsorption energy was calculated according to the equation as follows:

$$\Delta E_{ads} = E_{ads/slab} - (E_{ads} + E_{slab}). \tag{4}$$

where $E_{ads/slab}$, $E_{ads}$, and $E_{slab}$ are energies of the slab with adsorption, single adsorption, and the clean slab, respectively. The energy barriers of H$_2$O dissociation on the struck of Ru clusters were calculated via the Climbing-image nudged elastic band[55].

## Data availability

The data generated in this study are provided in Source Data file. Source data are provided with this paper.

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

## Acknowledgements

We appreciate the financial support of the National Natural Science Foundation (NNSF) of China (U21A20312, 21975162, 51902208) and the Shenzhen Government's Plan of Science and Technology (RCYX20200714114535052, JCYJ20170817095041212, and JCYJ20170818091657056). We also acknowledged the help of the Electron Microscopy Centre of Shenzhen University for testing the HADDF-STEM.

## Author contributions

C.H. conceived the project and idea. Q.H. designed the experiments, analyzed the data from these experiments, and wrote the manuscript. K.G., H.Z., J.C., L.M., and Q.H. carried out the synthesis, material characterizations, and electrochemical measurements. X.W. performed DFT calculations. H.Y. and J.L. commented on the manuscript.

## Competing interests

The authors declare no competing interests.

**Additional information**

