## [Peer Review File · Nature Communications]

Subnanometric Ru clusters with Upshifted d Band Center
Improve Performance for Alkaline Hydrogen Evolution
ReactionREVIEWER COMMENTS

Reviewer #1 (Remarks to the Author):

The reviewer has enjoyed reading this manuscript, well-written to explain the size effect of Ru clusters and their decisive role in HER performance. In detail, this work investigated the changes in size-dependent HER activity in alkaline media and suggested that the Ru cluster size governs the position of the d band center of Ru and consequent HER activity. The arguments have been underpinned by DFT calculations, spectroscopic analyses, and electrochemical methods. However, this reviewer raises some comments and questions, some of which may threaten the validity of their arguments and the general interest of the present work, which need to be clarified before accepting this manuscript.

[Comments and questions]

First, the Ru-x samples were synthesized with a biomass-derived carbon, upon which Ru³⁺ cations were impregnated and made clusters through the followed calcination process. However, as mentioned in the authors' previous works, the carbon substrate includes abundant N functionalities. Even if the functional groups are indispensable for stabilizing the Ru cations at the initial stage of catalyst synthesis steps, it is well known that the N-doped carbon can electronically interact with metal nanoparticles and consequently promotes their catalytic performance. Unfortunately, the DFT calculations and other discussions made with experimental results did not consider the possible metal-support interaction. Otherwise, to only elucidate the particle size effects, it would be better to use conventional carbon substrates. This reviewer recommends that authors check a previous work reported in [J. Am. Chem. Soc. 2013, 135, 8, 3073].

Second, this reviewer is very curious about the size effects of Ru clusters on their HER activities in acidic environments. Actually, all experiments and discussions were made in an alkaline electrolyte only. Still, as authors may know, one of the main issues in this society is unexpected HER activity changes depending on the electrolyte pH. The authors suggested the enhanced HER in the alkaline condition is attributed to the increment of the HBE, while entirely different trends can result in acidic conditions when one considers the recent consensus that the HBE is not the only determinant governing the HER activity. Furthermore, changes in the HBE can be experimentally verified using the ATR-SEIRAS analyses [J. Am. Chem. Soc. 142, 2020, 8748], which will strengthen the authors' arguments.

Third, the double-layer capacitance (C_{dl}) values reported in this manuscript were calculated by CV curves in a very narrow potential range from 0.05 VRHE to 0.07 VRHE. This experimental condition might be improper to measure the exact C_{dl} values because it can be overlapped with the HUPD region. Hence, the C_{dl} may need to be re-measured in an acceptable potential range at the so-called double layer region.

Fourth, the upshift of the d band center with the decreasing size of Ru clusters was supported by the experimental results of XPS and UPS, which manifests the changes in electronic structure. However, the deviations of the Ru 3p_{3/2} in the samples are seemingly low, and this needs to be shown more clearly. Also, this reviewer suggests that EXAFS and XANES spectra will be helpful to offer more direct evidence for the particle size and electronic structure changes, respectively.

[Minor comment]

First, the exact Ru contents on the catalysts should be provided (e.g., ICP-MS/OES data) as they could be varied during the calcination protocols.

Reviewer #2 (Remarks to the Author):

Exploring the metal size effect is an important topic in the research of electrocatalysis, however few has done work to study the metal size effect at a subnanometric scale. In this work, the authors prepared a series of Ru-based catalysts with different diameters and thus investigated the size-dependent activity of hydrogen evolution reaction (HER) at a subnanometric scale. By combining the results of theoretic calculations and experiments, they found that the d band center of Ru downshifts

in a nearly linear relationship with the increase of diameter, and the subnanometric Ru clusters with d band center closer to Fermi level display superior HER activity than SAs and larger nanoparticles. From both the perspective of catalyst performances and the reaction mechanistic understanding, I think this work can deliver a clear explanation and provide important guidelines for the design of robust electrocatalysts a subnanometric scale. To further improve the quality of this work, I have the following comments for the authors to reply and revise this manuscript.

1. In this work, the authors prepared subnanometric Ru clusters in diameter of ~ 1 nm by using porous carbon as a substrate. Why did the authors select a porous carbon as a substrate? If the carbon substrate is not porous, can the Ru clusters also be synthesized? Why are the obtained Ru clusters in diameter of ~ 1 nm? Can the authors prepare smaller Ru clusters?
2. With increasing the calcination temperature, the diameter of Ru was increased accordingly. Please explain why?
3. Why did the authors select the Ru clusters with 19, 55, and 79 atoms as structural models to perform the DFT calculations? It seemed that the size of ruthenium clusters was changed in an irregularly way.
4. The authors only calculated the work function values (Φ) of Ru in subnanometric Ru clusters and Ru nanoparticles through ultraviolet photoelectron spectroscopy (UPS). How about the Φ value of single atomic Ru? With Φ value of single atomic Ru, a clearer metal size effect at a subnanometric scale may be obtained.
5. The authors reported that the d band center of Ru downshifts in a nearly linear relationship with the increase of diameter. Is it a general trend for other transition metal? I leave this question as an open question. I would be appreciated if the authors can answer this question.
6. In Figure S20, it can be found that the gap of TOF values between subnanometric Ru clusters and single atomic Ru increases with the increase of current density. It seems not normal, please explain why?
7. In Figure 4a, the authors determined the water dissociation ability of different electrocatalysts via the value of *OH bind energy (OHBE). I am still not clear that why the value of OHBE can reflect the water dissociation ability. The authors should detailly explain it.
8. At the end of this manuscript, the authors loaded the subnanometric Ru clusters on the carbon paper with a large mass loading of 1 mg cm^{-2} to investigate its possibility as electrocatalysts for high-output industrial H₂ production, and they obtained a large current density of 1000 mA cm^{-2} at a small overpotential of 196 mV. It is well known that the hydrophilicity is very important for obtaining large current densities. How about the hydrophilicity of subnanometric Ru clusters?
9. The authors only compared the HER performance at a small current density of 10 mA cm^{-2} . It will be better if the authors also compare the HER performance at a large current density of 1000 mA cm^{-2} .
10. The conditions of electrochemical tests should be further specified. Did all electrochemical tests perform under some specified atmosphere? What is the scan rate to record the LSV curves?

Reviewer #3 (Remarks to the Author):

The paper discussed how the size-dependency of metal Ru affects the activity of the hydrogen evolution reaction. First of all, the introduction is demonstrated concisely and logically. The results are impressive by combining the evidence from both experimental and computational approaches. The investigation of size-dependent electrochemical activity provides a creative and visionary in the research field of searching for low-cost and high-activity electrochemical catalysts. Both the experimental and computational methods are solid and meet the expected standards, which results in outstanding results with compelling data analysis and presentations.

In addition, the author also extends the explorations from both the investigation of adding the second element and the feasibility of large-scale synthesis of Ru subnanometric clusters.

There are some typos (such as vs. instead of vs; theoretical instead of theoretic; systematic instead of systematical; on the basis of instead of on basis of; literature instead of literatures; most rapid instead of rapidest) and misuse of articles (the/a). Hope the authors could carefully go through the paper and correct those before publishing.

I want to think this is an excellent research paper to be published.

Response to Referees

Dear reviewers,

Thank you very much for your evaluation and comments on our manuscript, entitled “*Subnanometric Ru clusters with Upshifted d Band Center Enable Exceptional Performance for Alkaline Hydrogen Evolution Reaction*” (NCOMMS-22-09145). The manuscript has been revised accordingly and all of the changes were highlighted in the revised manuscript. We really like to thank all the reviewers for their valuable and constructive comments, which have made a much better presentation of the current study. The details of the revision are as follows:

Respond to reviewer 1

General comment: The reviewer has enjoyed reading this manuscript, well-written to explain the size effect of Ru clusters and their decisive role in HER performance. In detail, this work investigated the changes in size-dependent HER activity in alkaline media and suggested that the Ru cluster size governs the position of the d band center of Ru and consequent HER activity. The arguments have been underpinned by DFT calculations, spectroscopic analyses, and electrochemical methods. However, this reviewer raises some comments and questions, some of which may threaten the validity of their arguments and the general interest of the present work, which need to be clarified before accepting this manuscript

Reply: We greatly appreciate the reviewer’s positive comments on this work. Please see below our point-by-point responses to the reviewer’s concerns

Comment 1: First, the Ru-x samples were synthesized with a biomass-derived carbon, upon which Ru³⁺ cations were impregnated and made clusters through the followed calcination process. However, as mentioned in the authors’ previous works, the carbon substrate includes abundant N functionalities. Even if the functional groups are indispensable for stabilizing the Ru cations at the initial stage of catalyst synthesis steps,

it is well known that the N-doped carbon can electronically interact with metal nanoparticles and consequently promotes their catalytic performance. Unfortunately, the DFT calculations and other discussions made with experimental results did not consider the possible metal-support interaction. Otherwise, to only elucidate the particle size effects, it would be better to use conventional carbon substrates. This reviewer recommends that authors check a previous work reported in [J. Am. Chem. Soc. 2013, 135, 8, 3073]

Reply: We appreciate the reviewer for his/her thoughtful comments and questions. After we checked the previous literature (*J. Am. Chem. Soc. 2013, 135, 8, 3073*), we did agree with the reviewer's viewpoint that the N-doped carbon may electronically interact with metal nanoparticles to promote their catalytic performance. To this end, we synthesized porous carbon nanomaterials without nitrogen by using potassium citrate as precursors, and then we employed this porous carbon as substrates to obtain subnanometric Ru clusters in a diameter of ~1 nm. The resulting sample was denoted Ru-1.0/N-free C, and corresponding HRTEM images of Ru-1.0/N-free C further confirmed that the Ru clusters were in diameter of ~1 nm (Figure R1). As displayed in Figure R2, the HER activity of Ru-1.0/N-free C is inferior to the Ru-1.0 (supported on the porous N-doped carbon), suggesting that the nitrogen on the carbon may induce the electronic interaction between Ru clusters and carbon support for promoting the HER. The N-induced metal/support interactions were further investigated by performing DFT calculations using the structural models of Ru₁₉ clusters supported on the N-doped carbon (denoted Ru₁₉/NC) (Figure R3). The electronic density difference diagram indicates the presence of electron transfer from Ru₁₉ clusters (yellow region stands for the electron loss) to N-doped carbon (cyan region stands for the electron accumulation) at interfaces (Figure R4), thereby tuning the electronic structure of Ru. Bader charge analysis further suggests that the number of transferred electrons from Ru clusters to the nitrogen-doped carbon is 1.48 |e|. Notably, compared with pure Ru₁₉ (i.e., without the N-doped carbon support), the binding energy of the H* intermediate is decreased on the Ru₁₉/NC (Figure R5), which in turn facilitates the desorption of H* for boosting the HER. The optimized structural model of H* adsorbed on the Ru₁₉/NC is shown in

Figure R6. Moreover, the adsorption of Ru₁₉ clusters on N-doped carbon is a highly exothermic process with an adsorption energy of -8.15 eV, suggesting that the N-induced metal/support interaction can enhance the stability of Ru clusters during HER. Therefore, the N-induced metal/support interaction also contributes to the excellent HER activity of Ru-1.0 by optimizing the electronic structure of Ru and enhancing the stability of Ru.

According to the reviewer's advice, the related description was added as follows:

“As we previously reported⁴⁰, the porous carbon derived from the litchi shell contains a small amount of nitrogen (N) (1.1 wt%). The N elements in carbon may electronically interact with Ru clusters to promote their catalytic performance.⁴¹ To this end, we synthesized porous carbon nanomaterials without nitrogen by using potassium citrate as precursors (detailed synthetic process, please see Experimental section), and then we employed this porous carbon as substrates to obtain subnanometric Ru clusters in diameter of ~1 nm. The resulting sample was denoted Ru-1.0/N-free C, and corresponding HRTEM images of Ru-1.0/N-free C further confirmed that the Ru clusters were in diameter of ~1 nm. (Figure S20) As displayed in Figure S21, the HER activity of Ru-1.0/N-free C is inferior to the Ru-1.0 (supported on the porous N-doped carbon), suggesting that the nitrogen on the carbon may induce the electronic interaction between Ru clusters and carbon support for promoting the HER. The N-induced metal/support interactions were further investigated by performing DFT calculations using the structural models of Ru₁₉ clusters supported on the N-doped carbon (denoted Ru₁₉/NC) (Figure S22). The electronic density difference diagram indicates the presence of electron transfer from Ru₁₉ clusters (yellow region stands for the electron loss) to N-doped carbon (cyan region stands for the electron accumulation) at interfaces, thereby tuning the electronic structure of Ru (Figure S23). Bader charge analysis further suggests that the number of transferred electrons from Ru clusters to the nitrogen-doped carbon is 1.48 |e|. Notably, compared with pure Ru₁₉ (i.e., without the N-doped carbon support), the binding energy of the H* intermediate is decreased on the Ru₁₉/NC (Figure S24), which in turn facilitates the desorption of H* for boosting the HER. The optimized structural model of H* adsorbed on the Ru₁₉/NC is shown in

Figure S25. Moreover, the adsorption of Ru₁₉ clusters on N-doped carbon is a highly exothermic process with an adsorption energy of -8.15 eV, suggesting that the N-induced metal/support interaction can enhance the stability of Ru clusters during HER. Therefore, the N-induced metal/support interaction also contributes to the excellent HER activity of Ru-1.0 by optimizing the electronic structure of Ru and enhancing the stability of Ru.

”

Figure R1 TEM images of Ru-1.0/N-free C.

Figure R2 LSV curves of Ru-1.0/N-free C and Ru-1.0 for the HER in 1 M KOH.

Figure R3 Structural model of Ru₁₉/NC. The brown, ice blue, and silvery represent C, N, and Ru atoms.

Figure R4 Three-dimensional charge density difference diagrams of Ru₁₉/NC.

Figure R5 Comparing bind energies of hydrogen on the structural model of Ru₁₉/NC and Ru₁₉.

Figure R6 Optimized structure of hydrogen adsorbed on the Ru₁₉/NC. The brown, ice blue, silvery, and light pink represent C, N, Ru, and H atoms, respectively.

***Comment 2:** Second, this reviewer is very curious about the size effects of Ru clusters on their HER activities in acidic environments. Actually, all experiments and discussions were made in an alkaline electrolyte only. Still, as authors may know, one of the main issues in this society is unexpected HER activity changes depending on the electrolyte pH. The authors suggested the enhanced HER in the alkaline condition is attributed to the increment of the HBE, while entirely different trends can result in acidic conditions when one considers the recent consensus that the HBE is not the only determinant governing the HER activity. Furthermore, changes in the HBE can be experimentally verified using the ATR-SEIRAS analyses [J. Am. Chem. Soc. 142, 2020, 8748], which will strengthen the authors' arguments.*

Reply: We thank the reviewer for the insightful comments and suggestions. According to the reviewer's advice, we tested the HER performance of different electrocatalysts in the acid electrolyte (i.e., 0.05 M H₂SO₄ solution). Surprisingly, the HER activity trend in the acid electrolyte is totally opposite to that in alkaline, and the Ru-1.0 displayed the inferior HER activity than that of Ru-2.3 and Ru-3.1 in 0.05 M H₂SO₄ (Figure R7). It is widely accepted that the HER activity in the acid electrolyte is only determined by hydrogen binding energy (HBE) of electrocatalyst surface, while the alkaline HER activity is jointly determined by both water dissociation ability and (HBE). (*Angew. Chem. Int. Ed.* 2018, 57, 7568-7579; *J. Am. Chem. Soc.* 2020, 142, 8748-8754) In this work, DFT calculations and experiment results have demonstrated that the Ru-1.0 with upshifted d band center has too strong H adsorption strength and thus suppresses the

desorption of H, compared with the other two electrocatalysts. In this regard, the inferior HER activity in acid electrolytes than the other two electrocatalysts can be attributed to the too strong H adsorption strength. As for the alkaline HER, the water dissociation ability is a governing factor to control the HER activity, and the Ru-1.0 with stronger water dissociation ability displays superior HER activity than the other two electrocatalysts. The above finding further confirms that the superior HER activity of Ru-1.0 in alkaline is owing to the strong water dissociation ability.

According to the reviewer's advice, the related description was added as follows:

“HER activity of different electrocatalysts was also evaluated in the acid electrolyte (i.e., 0.05 M H₂SO₄ solution). (Figure S32) Surprisingly, the HER activity trend in the acid electrolyte is totally opposite to that in alkaline, and the Ru-1.0 displayed the inferior HER activity than that of Ru-2.3 and Ru-3.1 in 0.05 M H₂SO₄. It is widely accepted that the HER activity in the acid electrolyte is only determined by hydrogen binding energy (HBE) of the electrocatalyst surface, while the alkaline HER activity is jointly determined by both water dissociation ability and (HBE).^{11, 49-50} In this work, DFT calculations and experiment results have demonstrated that the Ru-1.0 with upshifted d band center has too strong H adsorption strength and thus suppresses the desorption of H, compared with the other two electrocatalysts. In this regard, the inferior HER activity in acid electrolytes than the other two electrocatalysts can be attributed to the too strong H adsorption strength. As for the alkaline HER, the water dissociation ability is a governing factor to control the HER activity, and the Ru-1.0 with stronger water dissociation ability displays superior HER activity than the other two electrocatalysts. The above finding further confirms that the superior HER activity of Ru-1.0 in alkaline is owing to the strong water dissociation ability.”

Figure R7 LSV curves of Ru-1.0, Ru-2.3, and Ru-3.1 for the HER in the electrolyte of 0.05 M H₂SO₄ solution.

Comment 3: Third, the double-layer capacitance (C_{dl}) values reported in this manuscript were calculated by CV curves in a very narrow potential range from 0.05 VRHE to 0.07 VRHE. This experimental condition might be improper to measure the exact C_{dl} values because it can be overlapped with the HUPD region. Hence, the C_{dl} may need to be re-measured in an acceptable potential range at the so-called double layer region.

Reply: We thank the reviewer for the valuable comments and suggestions. According to the reviewer's advice, we have retested the C_{dl} at a potential range from 0.21 to 0.31 V vs. RHE (Please see Figure R8)

Figure R8 CV of (a) Ru-1.0, (b) Ru-2.3, and (c) Ru-3.1 at different scan rates over a potential range of 0.05-0.07 V vs RHE. (d) Profiles of capacitance Δj ($|j_{\text{charge}} - j_{\text{discharge}}|$) as a function of scan rates.

Comment 4: Fourth, the upshift of the *d* band center with the decreasing size of Ru clusters was supported by the experimental results of XPS and UPS, which manifests the changes in electronic structure. However, the deviations of the Ru 3p3/2 in the samples are seemingly low, and this needs to be shown more clearly. Also, this reviewer suggests that EXAFS and XANES spectra will be helpful to offer more direct evidence for the particle size and electronic structure changes, respectively.

Reply: We do agree with the reviewer's viewpoint that it will be helpful to offer more direct evidence for the electronic structure changes by measuring EXAFS and XANES spectra. As soon as the reviewer's comment was sent to us, we connected with "Shanghai Synchrotron Radiation Facility" to apply for conducting XAFS and XANES. However, Shanghai in China is still suffering from serious COVID-19, and the "Shanghai Synchrotron Radiation Facility" is being closed now. As such, we can not perform XAFS and XANES timely. On the other hand, the UPS is a powerful tool to investigate the surface electronic structure and work function values of metal surface. In most cases, the results of UPS are in good accordance with the XAFS and XANES.

Moreover, our DFT calculations are in consistence with the UPS results, jointly confirming the trend of the d band center shift induced by the particle size. In the future, we will perform XAFS and XANES spectra on different electrocatalysts, and we believe that the obtained results will be in good accordance with our results of UPS and DFT calculations.

Comment 5: First, the exact Ru contents on the catalysts should be provided (e.g., ICP-MS/OES data) as they could be varied during the calcination protocols.

Reply: We thank the reviewer for the good suggestion. According to the reviewer's advice, we tested the content of Ru on different electrocatalysts via ICP-AES. As displayed in Figure R9, the content of Ru in all three electrocatalysts is quite similar, and only a slight decrease in Ru content can be observed with the increase of calcination temperature. According to the reviewer's advice, the related description was added as follows:

“In order to eliminate the influence of Ru content on the electrocatalytic performance, all samples were prepared using the same amount of Ru source, and the mass loading of Ru was determined by using inductively coupled plasma-atomic emission spectrometry (ICP-AES). The results indicate that the Ru content in all three samples is quite similar (~2.0 wt%), and only a slight decrease in Ru content can be observed with the increase of calcination temperature. (Figure S7)”

Figure R9 Comparing the Ru content on the Ru-1.0, Ru-2.3, and Ru-3.1.

Respond to reviewer 2

*General comment: Exploring the metal size effect is an important topic in the research of electrocatalysis, however few has done work to study the metal size effect at a subnanometric scale. In this work, the authors prepared a series of Ru-based catalysts with different diameters and thus investigated the size-dependent activity of hydrogen evolution reaction (HER) at a subnanometric scale. By combining the results of theoretic calculations and experiments, they found that the *d* band center of Ru downshifts in a nearly linear relationship with the increase of diameter, and the subnanometric Ru clusters with *d* band center closer to *Fermi* level display superior HER activity than SAs and larger nanoparticles. From both the perspective of catalyst performances and the reaction mechanistic understanding, I think this work can deliver a clear explanation and provide important guidelines for the design of robust electrocatalysts a subnanometric scale. To further improve the quality of this work, I have the following comments for the authors to reply and revise this manuscript.*

Reply: We greatly appreciate the reviewer's high praise for exploring the metal-size effect at a subnanometric scale.

***Comment 1:** In this work, the authors prepared subnanometric Ru clusters in diameter of ~1 nm by using porous carbon as a substrate. Why did the authors select a porous carbon as a substrate? If the carbon substrate is not porous, can the Ru clusters also be synthesized? Why are the obtained Ru clusters in diameter of ~1 nm? Can the authors prepare smaller Ru clusters?*

Reply: We thank the reviewer for the useful comment. First, the goal that we selected the porous carbon as the substrate is to utilize the spatial confinement effect of micropores for generating subnanometric Ru clusters. As displayed in Figure R10, after loading the Ru cluster on the surface of porous carbon, the surface area of the carbon substrate decreases from 1890 to 1630 m² g⁻¹, and the intensity of micropore distribution is reduced, implying that the spatial confinement of micropores may be essential for the formation of subnanometric clusters in the Ru-1.0. Second, if the carbon substrate is not porous, larger Ru nanoparticles will be generated (Figure R11), further confirming the important role of micropores in generating Ru clusters. Third, it is still challenging work for us to prepare Ru clusters less than 1 nm because we can not prepare porous carbon nanomaterials with pore diameter less than 1 nm for confining the growth of Ru clusters.

According to the reviewer's advice, the related description was added as follows:

“In addition, after loading Ru clusters, the surface area of carbon substrate decreases from 1890 to 1630 m² g⁻¹, and the intensity of micropore distribution is reduced (Figure S13), implying that the spatial confinement of micropores may be essential for the formation of subnanometric clusters in the Ru-1.0. If the carbon substrate is not porous, larger Ru nanoparticles are generated (Figure S14), further confirming the important role of micropores in generating Ru clusters.”

Figure R10 (a) N₂ adsorption-desorption isotherms of porous carbon and Ru-1.0, (b) corresponding pore distribution of above noted two samples based on a method of BJH.

Figure R11 TEM images of the sample prepared by using non-porous carbon as substrates.

Comment 2: With increasing the calcination temperature, the diameter of Ru was increased accordingly. Please explain why?

Reply: We thank the reviewer for the comment. The increase of Ru diameter is because the higher calcination temperature accelerates the movement of Ru atoms during the calcination and thus leads to the aggregation of Ru atoms. According to the reviewer's advice, the related description was added as follows:

“Intriguingly, with the increase of calcination temperature from 300, 400 to 500 °C, the diameter of Ru crystals increases accordingly from 1, 2.3, to 3.1 nm. This is because that the higher calcination temperature accelerates the movement of Ru atoms during the calcination and thus leads to the growth of Ru crystals.”

Comment 3: Why did the authors select the Ru clusters with 19, 55, and 79 atoms as structural models to perform the DFT calculations? It seemed that the size of ruthenium clusters was changed in an irregularly way.

Reply: We thank the reviewer for the thoughtful comment. We selected odd-number of Ru_n clusters ($n = 19, 55, \text{ and } 79$) to investigate the size effect of Ru on HER activity because they possess singly occupied highest occupied molecular orbitals (HOMO), rendering them more active than the even-numbered Ru clusters with a doubly occupied HOMO. (*Comput. Theor. Chem.* 2013, 1009, 8–16) According to the reviewer’s advice, the related description was added as follows:

“We selected odd-number of Ru_n clusters ($n = 19, 55, \text{ and } 79$) to investigate the size effect of Ru on HER activity because they possess singly occupied highest occupied molecular orbitals (HOMO), rendering they more active than the even-numbered Ru clusters with a doubly occupied HOMO.³²”

Comment 4: The authors only calculated the work function values (Φ) of Ru in subnanometric Ru clusters and Ru nanoparticles through ultraviolet photoelectron spectroscopy (UPS). How about the Φ value of single atomic Ru? With Φ value of single atomic Ru, a clearer metal size effect at a subnanometric scale may be obtained.

Reply: We thank the reviewer for the thoughtful suggestion. According to the reviewer’s advice, we determined the Φ value of Ru SAs through UPS spectra. (Figure R12) It can be found that the Φ value of Ru SAs (3.12 eV) is smaller than the Ru-1.0 (3.32 eV), implying that the higher Fermi energy level of Ru SAs than the Ru-1.0. This further confirms the fact that the d band center of Ru upshifts with the decrease of Ru particle size. The related description was added as follows:

“Moreover, we also determined the Φ value of Ru SAs through UPS spectra.

(Figure S18) It can be found that the Φ value of Ru SAs (3.12 eV) is smaller than the Ru-1.0 (3.32 eV), implying that the higher Fermi energy level of Ru SAs than the Ru-1.0. This further confirms the fact that the d band center of Ru upshifts with the decrease of Ru particle size.”.

Figure R12 UPS spectra of Ru SAs.

***Comment 5:** The authors reported that the d band center of Ru downshifts in a nearly linear relationship with the increase of diameter. Is it a general trend for other transition metal? I leave this question as an open question. I would be appreciated if the authors can answer this question.*

Reply: We thank the reviewer for the comment. First, enormous DFT calculations have fully demonstrated that the d band center of Cu downshifts with the increase of diameter. Second, in our previous literature, we compared the activity of larger Cu nanoparticles (NPs) (i.e., in diameter of 19 nm) and subnanometric Cu clusters for the carbon dioxide reduction (CO₂RR). We found that Cu clusters with upshifted d center have stronger adsorption strength of *CO and *H for suppressing hydrogen evolution reaction (HER) and promoting the generation of CH₄, compared with much bigger NPs (~19 nm). Thus, we believe that it may be a general trend for other transition metals.

Comment 6: In Figure S20, it can be found that the gap of TOF values between subnanometric Ru clusters and single atomic Ru increases with the increase of current density. It seems not normal, please explain why?

Reply: We thank the reviewer for the useful comment. The increased TOF gap between subnanometric Ru clusters and single atomic Ru can be due to the stronger dissociation ability of Ru clusters for increasing the H* coverage at the electrocatalyst surface, thereby achieving high TOF values at small overpotentials.

“Intriguingly, the gap of TOF values between Ru clusters and SAs Ru increase with the increase of overpotentials. This can be due to the stronger dissociation ability of Ru clusters for increasing the H* coverage at the electrocatalyst surface, thereby achieving high TOF values at small overpotentials.”

*Comment 7: In Figure 4a, the authors determined the water dissociation ability of different electrocatalysts via the value of *OH bind energy (OHBE). I am still not clear that why the value of OHBE can reflect the water dissociation ability. The authors should detailly explain it.*

Reply: We thank the reviewer for the important comment. Previously, Qiao's group utilized operando Raman spectra to investigate the influence of OH⁻ concentration in electrolytes on the HER activity of commercial Pt/C electrocatalysts. They found that the OH⁻ concentration greatly impacted the HER activity, and a high OH⁻ concentration could significantly boost the water dissociation to provide a unique acid-like local reaction environment for enhancing HER activity. Therefore, the adsorption ability of *OH is a key parameter to evaluate the water dissociation ability of an electrocatalyst, and the water dissociation ability of different electrocatalysts can be determined by the *OH bind energy (OHBE). The related description was added as follows:

“As demonstrated by Qiao's group, the OH⁻ concentration greatly impacted the HER activity of Pt/C electrocatalyst, and a high OH⁻ concentration could significantly boost the water dissociation to provide a unique acid-like local reaction environment for enhancing HER activity. Thus, the adsorption ability of *OH is a key parameter to evaluate the water dissociation ability of an electrocatalyst, and the water dissociation

ability of different electrocatalysts can be determined by the *OH bind energy (OHBE)^{44, 45}”

Comment 8: *At the end of this manuscript, the authors loaded the subnanometric Ru clusters on the carbon paper with a large mass loading of 1 mg cm⁻² to investigate its possibility as electrocatalysts for high-output industrial H₂ production, and they obtained a large current density of 1000 mA cm⁻² at a small overpotential of 196 mV. It is well known that the hydrophilicity is very important for obtaining large current densities. How about the hydrophilicity of subnanometric Ru clusters?*

Reply: We thank the reviewer for the comment. According to the reviewer’s advice, we tested the water contact of the Ru-1.0 sample, and the results indicated that the Ru-1.0 had good hydrophilicity with a water contact of merely 15.2°. (Figure R12) Good hydrophilicity promotes the water adsorption and thus enables rapid mass transport to yield large current densities. The related description was added as follows:

“The excellent performance of Ru-1.0 at large current densities may be attributed to the following two factors: 1) the strong water dissociation ability of Ru-1.0 enables rapid water dissociation for increasing the coverage of H* and thus promoting the dimerization of H* at a high rate; 2) the porous carbon support with good hydrophilicity (Figure S37) and high surface area (i.e., 1890 m² g⁻¹) significantly accelerates the mass transport during HER.”

Figure R12 Water contact angle of Ru-1.0.

Comment 9: *The authors only compared the HER performance at a small current*

density of 10 mA cm⁻². It will be better if the authors also compare the HER performance at a large current density of 1000 mA cm⁻².

Reply: We thank the reviewer for the comment. According to the reviewer's advice, we compared the HER activity of Ru-1.0 and other reported electrocatalysts at the current density of 1000 mA cm⁻² (Table R1) It can be found that the Ru-1.0 compares favorably with other reported electrocatalysts. The related description was added as follows:

“Notably, the Ru-1.0 also compares favorably with other reported electrocatalysts at the current density of 1000 mA cm⁻² (Table S3).”

Table R1 Comparing the HER activity of our Ru-1.0 with other reported electrocatalysts on basis of overpotentials at the current density of 1000 mA cm⁻².

Catalysts	Overpotential (mV)	Reference
Ru-1.0	196	This work
HC-MoS ₂ /Mo ₂ C	412	Nat. Commun. 2020, 11, 3724
NiMoO _x /NiMoS	236	Nat. Commun. 2020, 11, 5462
Co(10.4)/Se-MoS ₂ -NF	382	Nat. Commun. 2020, 11, 3315
Co-Mo ₅ N ₆	280	Adv. Energy Mater. 2020, 10, 2002176
N ₂ P/NF	306	J. Am. Chem. Soc. 2019, 141, 7537-7543
FeP/Ni ₂ P	293	Nat. Commun. 2018, 9, 2551

Comment 10: The conditions of electrochemical tests should be further specified. Did all electrochemical tests perform under some specified atmosphere? What is the scan rate to record the LSV curves?

Reply: We thank the reviewer for the comment. According to the reviewer's advice, the conditions of electrochemical tests have been further specified. (Please see “Experimental Section”)

Respond to reviewer 3

General comment: The paper discussed how the size-dependency of metal Ru affects the activity of the hydrogen evolution reaction. First of all, the introduction is demonstrated concisely and logically. The results are impressive by combining the evidence from both experimental and computational approaches. The investigation of size-dependent electrochemical activity provides a creative and visionary in the research field of searching for low-cost and high-activity electrochemical catalysts. Both the experimental and computational methods are solid and meet the expected standards, which results in outstanding results with compelling data analysis and presentations.

In addition, the author also extends the explorations from both the investigation of adding the second element and the feasibility of large-scale synthesis of Ru subnanometric clusters.

There are some typos (such as vs. instead of vs; theoretical instead of theoretic; systematic instead of systematical; on the basis of instead of on basis of; literature instead of literatures; most rapid instead of rapidest) and misuse of articles (the/a). Hope the authors could carefully go through the paper and correct those before publishing.

I want to think this is an excellent research paper to be published.

Reply: We are very grateful for the reviewer's highly positive assessment of our work. According to the reviewer's advice, we have rechecked the manuscript and revised all typos in this manuscript.

We would like to use this chance to thank all the referees very much for editing and reviewing this manuscript. Some of constructive suggestions have made a real improvement of this manuscript. We truly appreciate for your help.

Best regards,

Chuanxin He

REVIEWERS' COMMENTS

Reviewer #1 (Remarks to the Author):

The authors deeply discussed all comments raised by this reviewer, and the manuscript was revised well with additional data, which corroborates and strengthens the authors' findings. Now, this reviewer can recommend its publication in the journal of Nature Communications.

Reviewer #2 (Remarks to the Author):

Authors responded all the reviewers's comments well.
I also satisfied with the reviewers' responses.